# Senescent Human Pancreatic Stellate Cells Secrete CXCR2 Agonist CXCLs to Promote Proliferation and Migration of Human Pancreatic Cancer AsPC-1 and MIAPaCa-2 Cell Lines

**DOI:** 10.3390/ijms23169275

**Published:** 2022-08-17

**Authors:** Tetsuya Takikawa, Shin Hamada, Ryotaro Matsumoto, Yu Tanaka, Fumiya Kataoka, Akira Sasaki, Atsushi Masamune

**Affiliations:** Division of Gastroenterology, Tohoku University Graduate School of Medicine, Sendai 980-8574, Japan

**Keywords:** cancer-associated fibroblast, cellular senescence, chemokine, myofibroblast, pancreatic ductal adenocarcinoma, senescence-associated secretory phenotype, tumor microenvironment

## Abstract

Interactions between pancreatic cancer cells and pancreatic stellate cells (PSCs) play an important role in the progression of pancreatic cancer. Recent studies have shown that cellular senescence and senescence-associated secretory phenotype factors play roles in the progression of cancer. This study aimed to clarify the effects of senescence-induced PSCs on pancreatic cancer cells. Senescence was induced in primary-cultured human PSCs (hPSCs) through treatment with hydrogen peroxide or gemcitabine. Microarray and Gene Ontology analyses showed the alterations in genes and pathways related to cellular senescence and senescence-associated secretory phenotype factors, including the upregulation of C-X-C motif chemokine ligand (CXCL)-1, CXCL2, and CXCL3 through the induction of senescence in hPSCs. Conditioned media of senescent hPSCs increased the proliferation—as found in an assessment with a BrdU incorporation assay—and migration—as found in an assessment with wound-healing and two-chamber assays—of pancreatic cancer AsPC-1 and MIAPaca-2 cell lines. SB225002, a selective CXCR2 antagonist, and SCH-527123, a CXCR1/CXCR2 antagonist, attenuated the effects of conditioned media of senescent hPSCs on the proliferation and migration of pancreatic cancer cells. These results suggest a role of CXCLs as senescence-associated secretory phenotype factors in the interaction between senescent hPSCs and pancreatic cancer cells. Senescent PSCs might be novel therapeutic targets for pancreatic cancer.

## 1. Introduction

Pancreatic cancer is the most common type of pancreatic malignancy, and its incidence and mortality rates are increasing worldwide [1]. The American Cancer Society estimates that 62,210 patients will be diagnosed and 49,830 will die of pancreatic cancer in 2022, making it the third leading cause of cancer-related deaths in the United States [2]. Pancreatic cancer is an intractable disease because of the difficulty in early diagnosis and its limited response to chemotherapy and radiotherapy, with 5-year survival rates as low as approximately 10% in the United States and Japan [3,4].

Pancreatic cancer is characterized by abundant desmoplastic stroma, accounting for up to 90% of the tumor volume [5,6]. The desmoplastic stroma, which are composed of the extracellular matrix (ECM), various types of cells, and tumor vessels, forms both supporting and restraining tumor microenvironments for pancreatic cancer [7,8]. Pancreatic stellate cells (PSCs), which were first identified and characterized in 1998, play a pivotal role in the development of desmoplastic stroma by producing large amounts of ECM proteins, such as collagen and fibronectin [5,9,10]. Moreover, PSCs directly promote cell proliferation and invasion, the epithelial–mesenchymal transition, and drug resistance in pancreatic cancer cells through the actions of cytokines, growth factors, and exosomes [5,11,12,13,14]. PSCs and desmoplastic stroma, the major components of the tumor microenvironment in pancreatic cancer, have attracted attention as therapeutic targets due to their cancer-promoting effects [7]. However, the inhibitory effects of PSCs in pancreatic cancer have also been suggested over the last decade. The depletion of activated PSCs and desmoplastic stroma in genetically engineered mice led to the progression of pancreatic cancer and decreased survival time [15,16]. Clinical trials of hedgehogs and smoothened inhibitors targeting desmoplastic stroma also failed to show a treatment benefit [17,18]. These findings indicate the existence of cancer-restraining PSCs and suggest that various phenotypes of PSCs are intermingled in the tumor microenvironment of pancreatic cancer. The development of novel therapies targeting PSCs requires consideration of their heterogeneity [6,14].

Cellular senescence was originally observed as a phenomenon in which normal cells cease proliferation after repeated division [19,20]. Subsequent studies have shown that numerous stressors, such as radiation, genotoxic drugs, and oxidative stress, induce cellular senescence [20,21]. Recently, it was shown that senescent cells secrete various cytokines, chemokines, and growth factors, which are collectively known as the senescence-associated secretory phenotype (SASP) [20,21,22]. Although cellular senescence is considered a physiological phenomenon that maintains homeostasis against various stresses, SASP factors derived from senescent cells comprising the tumor microenvironment facilitate cancer progression [22,23,24,25,26]. Moreover, the removal of SASP factors by selectively killing senescent cells inhibits cancer progression [24,27]. Although therapeutic agents targeting senescent cells—so-called senolytic drugs—are expected to be novel cancer therapeutics, the effects of senescence and SASP factors of PSCs on pancreatic cancer cells remain unclear.

To address this issue, we induced senescence in primary-cultured human PSCs (hPSCs) and investigated the effects of SASP factors in conditioned media of senescence-induced hPSCs on pancreatic cancer cells.

## 2. Results

### 2.1. Senescence Induction in hPSCs using H_2_O_2_ and Gemcitabine

In this study, we used hydrogen peroxide (H_2_O_2_) and gemcitabine, a standard drug for pancreatic cancer treatment, for the induction of senescence in hPSCs. The H_2_O_2_ and gemcitabine treatment decreased the proliferation of hPSCs in a dose-dependent manner (Figure 1A). The H_2_O_2_ and gemcitabine treatment increased the expression of cyclin-dependent kinase inhibitor 1A (*CDKN1A/*p21) at both the mRNA and protein levels (Figure 1B–D), while the expression of tumor protein p53 (*TP53/*p53) was only increased at the protein level (Figure 1D). Although the *TP53/*p53 mRNA expression levels did not change (Figure 1B,C), an increase in the level of *MDM2*, a key molecule in the negative feedback loop of p53, was observed (Figure 1B,C). A senescence-associated β-galactosidase (SA-β-gal) assay showed that the H_2_O_2_ and gemcitabine treatment increased the number of SA-β-gal-positive cells in a dose-dependent manner (Figure 1E,F).

### 2.2. Conditioned Media of Senescence-Induced hPSCs Increased the Proliferation and Migration of Pancreatic Cancer Cells

To clarify the effects of senescence-induced hPSCs on pancreatic cancer cells, we prepared conditioned media (CM) of H_2_O_2_-treated hPSCs (H_2_O_2_ PSC-CM), gemcitabine-treated hPSCs (GEM PSC-CM), and untreated hPSCs (Ctrl PSC-CM). We first assessed the effects of PSC-CM on the proliferation of human pancreatic cancer cell lines (AsPC-1 and MIAPaCa-2) with a bromodeoxyuridine (BrdU) incorporation assay. Although Ctrl PSC-CM increased the proliferation of AsPC-1 and MIAPaCa2 cells, both H_2_O_2_ PSC-CM and GEM PSC-CM increased proliferation more strongly than Ctrl PSC-CM (Figure 2A,B). Next, we assessed migration through wound-healing and two-chamber assays. H_2_O_2_ PSC-CM and GEM PSC-CM increased the migration of AsPC-1 and MIAPaCa2 cells toward the wound area more strongly than Ctrl PSC-CM (Figure 2C,D). Similar results were obtained for the two-chamber assay (Figure 2E,F). These results suggested that senescent hPSCs increased the proliferation and migration of pancreatic cancer cells.

### 2.3. Senescence Induction Altered Gene Expression Profiles in hPSCs

To provide an overview of differentially expressed genes in senescence-induced hPSCs, we compared the gene expression profiles between untreated and H_2_O_2_-treated hPSCs by using Agilent’s microarray. The microarray analysis identified 565 upregulated and 605 downregulated probes following the induction of senescence in hPSCs (Figure 3A). The downregulated genes included actin alpha 2 (*ACTA2*/α-smooth muscle actin [SMA]), a marker of activated PSCs, and ECM components, such as collagen type I alpha 1 chain (*COL1A1*), collagen type IV alpha 1 chain (*COL4A1*), and collagen type V alpha 1 chain (*COL5A1*). The decreased expression of these genes was validated with quantitative real-time PCR (Figure 3B). We performed Gene Ontology (GO) and Kyoto Encyclopedia of Genes and Genomes (KEGG) pathway enrichment analyses to elucidate the biological roles and signaling pathways involved in senescence induction in hPSCs. Table 1 shows the top five enriched GO terms for biological processes, cellular components, and molecular functions. Enriched genes included those involved in the cell cycle, cell division, DNA replication, growth factor, and cytokine, which are known to be related to cellular senescence and SASP. The top 10 enriched KEGG pathways included those related to senescence and SASP, such as the cell cycle, *p53* signaling pathway, focal adhesion, cytokine–cytokine receptor interaction, and ECM–receptor interaction (Table 2).

We further compared the gene sets of senescence-induced hPSCs with those of senescence-induced human hepatic stellate cells (hHSCs) by using gene set enrichment analysis (GSEA). Both of upregulated and downregulated genes in the induction of senescence in hPSCs were strongly correlated with those in hHSCs (both *P* < 0.01; false-discovery rate: *q* < 0.01) (Figure 3C).

### 2.4. CXCL1, CXCL2, and CXCL3 Acted as SASP Factors in Senescence-Induced hPSCs

The microarray analysis showed that the levels of various cytokines, chemokines, and growth factors were increased in senescence-induced hPSCs. The gene list involved in GO terms “cytokine”, “growth factor”, and “secreted”, which may be related to SASP factors, is shown in Appendix A. Among them, we focused on C-X-C motif chemokine ligand (*CXCL*)-1 (6.1-fold), *CXCL2* (2.2-fold), and *CXCL3* (6.6-fold) as SASP factors, as they act on the common receptor, C-X-C motif chemokine receptor 2 (CXCR2), and the CXCLs/CXCR2 axis in the tumor–stromal interaction promotes the progression of pancreatic cancer. The increased mRNA levels of *CXCL1*, *CXCL2,* and *CXCL3* in senescence-induced hPSCs were validated through quantitative real-time PCR (Figure 4A,B). An enzyme-linked immunosorbent assay (ELISA) showed increased CXCL1 concentration in the conditioned media of hPSCs after senescence induction with H_2_O_2_ or gemcitabine (Figure 4C).

### 2.5. Inhibition of CXCR1/CXCR2 Attenuated the Cancer-Promoting Effects of Senescence-Induced hPSCs

To clarify the roles of CXCLs in the cancer-promoting effects of senescence-induced hPSCs, we used SB225002, a selective CXCR2 antagonist, and SCH-527123, a CXCR1/CXCR2 antagonist, to block the CXCLs/CXCR2 axis. SB225002 and SCH-527123 inhibited the proliferation of AsPC-1 and MIAPaca-2 cells induced by H_2_O_2_ PSC-CM in a dose-dependent manner (Figure 5A). Similarly, SB225002 and SCH-527123 inhibited the proliferation of AsPC-1 and MIAPaca-2 cells induced by GEM PSC-CM in a dose-dependent manner (Figure 5B). SB225002 and SCH-527123 attenuated the stimulatory effects of H_2_O_2_ PSC-CM and GEM PSC-CM on the migration of AsPC-1 and MIAPaca-2 cells, as found in the assessment with the wound-healing assay (Figure 5C,D) and the two-chamber assay (Figure 5E,F). These results suggest that CXCLs act as SASP factors of senescence-induced hPSCs, promoting the proliferation and migration of pancreatic cancer cells.

## 3. Discussion

In this study, we investigated the effects of senescent hPSCs on pancreatic cancer cells. We showed that senescent hPSCs promoted the proliferation and migration of the pancreatic cancer AsPC-1 and MIAPaCa-2 cell lines. Through microarray and GO analyses, we found that senescence induction resulted in the alteration of gene expression and pathways in a similar manner to other cell types [28,29,30,31,32]. Furthermore, we identified CXCL1, CXCL2, and CXCL3 as SASP factors secreted by senescence-induced hPSCs, and the inhibition of the CXCLs/CXCR2 axis attenuated their stimulating effects on the proliferation and migration of pancreatic cancer cells. These results suggest that senescent hPSCs in the tumor microenvironment of pancreatic cancer exert cancer-promoting effects and may serve as novel therapeutic targets for pancreatic cancer.

Cellular senescence and SASP play important roles in both tumor-promoting and tumor-suppressing interactions between tumors and their microenvironment [33]. Although the expression of senescence markers and SASP factors has been reported in pancreatic cancer cells and stromal cells [34,35], the roles of senescence and SASP of hPSCs in pancreatic cancer remain unknown. We here induced senescence in hPSCs using H_2_O_2_, a reactive oxygen species (ROS) [32], and gemcitabine. In pancreatic cancer tissues that are characterized by abundant desmoplastic stroma, hypoxic and nutrient-limited environments lead to ROS accumulation [36]. Elevated ROS levels in pancreatic cancer cells and PSCs exert cancer-promoting effects [37,38]. hPSCs treated with H_2_O_2_ or gemcitabine exerted similar cancer-promoting effects. Along with our results, Toste et al. [39] reported that gemcitabine-resistant cancer-associated fibroblasts showed increased expression of various inflammatory mediators and stimulated migration of pancreatic cancer cells. Gemcitabine treatment induces senescence in pancreatic cancer cells and increases CXCL8 expression [40]. These results suggest that cellular senescence and SASP factors in PSCs that are induced by oxidative stress and treatment of pancreatic cancer have cancer-promoting effects.

To the best of our knowledge, this is the first comprehensive gene expression analysis of senescent hPSCs. Induction of senescence in hPSCs revealed enriched GO terms involved in the cell cycle, cell division, DNA replication, growth factors, and cytokines, which are known to be related to cellular senescence and SASP in other cell types [28,29,30]. KEGG analysis revealed that differentially expressed genes were involved in senescence and SASP-related pathways (cell cycle, *p53* signaling pathway, focal adhesion, cytokine–cytokine receptor interaction, and ECM–receptor interaction) [29,30,31]. Because PSCs are regarded as counterparts of HSCs, a major player in the development of liver fibrosis in the pancreas [41,42], we compared the senescence-induced alterations in gene expression between hPSCs and hHSCs. Although recent studies showed morphological and functional differences between PSCs and HSCs [43,44], GSEA analysis showed that both upregulated and downregulated genes in senescence induction in hPSCs were strongly correlated with those in hHSCs [29]. These results suggest similar phenotypic alterations through senescence induction in PSCs and HSCs.

We identified CXCL1, CXCL2, and CXCL3 as SASP factors secreted by senescent hPSCs. These CXCLs, which act on the common receptor CXCR2, are members of the inflammatory chemokine family, and they have similar structures and play important roles in various malignancies, including pancreatic cancer [45,46]. SB225002, a selective CXCR2 antagonist [47], and SCH-527123, a CXCR1/CXCR2 antagonist [48], canceled the cancer-promoting effects of conditioned media from senescent hPSCs. Lian et al. [49] reported that high CXCL1 expression levels in the stroma are correlated with poor prognosis in patients with pancreatic cancer. Sano et al. [46] reported that inhibition of the CXCLs/CXCR2 axis in tumor–stromal interactions contributes to improved prognosis in a mouse model of pancreatic cancer. Thus, cellular senescence may play a role in the CXCLs/CXCR2 axis in pancreatic cancer.

We found that senescence induction in hPSCs decreased the expression levels of *ACTA2* (α-SMA) and ECM components, such as *COL1A1*, *COL4A1*, and *COL5A1*. The microarray analysis showed increased levels of matrix metalloproteinases (MMPs), such as *MMP1*, *MMP3*, and *MMP12*, which break down the ECM components (data not shown). These data suggest that senescence induction in PSCs results in anti-fibrotic properties that are similar to those reported in HSCs [29]. Despite their anti-fibrotic properties, senescent hPSCs exerted cancer-promoting effects, such as stimulating proliferation and migration of pancreatic cancer cells. In a normal pancreas, PSCs are quiescent. In response to pancreatic inflammation and injury, quiescent PSCs become activated PSCs, accompanied by increased α-SMA expression and production of ECM proteins, cytokines, and growth factors [5]. Previous studies have shown the cancer-promoting roles of PSCs [5]. For example, interleukin-6 secreted from PSCs promotes cell proliferation, stemness, and drug resistance of pancreatic cancer cells [14]. Xu et al. [50] reported that PSCs promote the epithelial–mesenchymal transition and drug resistance of pancreatic cancer cells via paracrine hepatocyte growth factor. In addition, PSCs are known to interact with cells other than pancreatic cancer cells. A co-culture of PSCs and pancreatic acinar cells inhibited the increase in cytosolic calcium and release of amylase in pancreatic acinar cells [51,52]. Kikuta et al. [53] reported that a co-culture of PSCs and pancreatic β cells reduced insulin expression and induced apoptosis in pancreatic β cells. Importantly, the inhibitory effects of PSCs on pancreatic cancer have been reported [15,16]. The development of therapies targeting PSCs requires consideration of their heterogeneity [6,14]. Our study implies that senescent PSCs, which secrete various SASP factors despite having low α-SMA expression and anti-fibrotic effects, have features that are different from those of quiescent and activated PSCs. Further studies are required to characterize the properties of senescent PSCs in greater detail.

This study has several limitations. First, although we revealed the involvement of CXCLs, as SASP factors secreted by senescent hPSCs, in pancreatic cancer progression, the outcomes of senescence inhibition by itself remain unclear. Microarray analysis showed that the levels of several cytokines and growth factors other than CXCLs were also upregulated through senescence induction in hPSCs; therefore, the inhibition of senescence itself would more strongly restrain the cancer-promoting effects in pancreatic cancer, as with other organ malignancies [22,23,24,25,26]. Second, this study used primary-cultured hPSCs, and different primary-cultured hPSCs might have different phenotypes and different effects on pancreatic cancer cells [54]. Lastly, this study was conducted only in vitro; therefore, further in vivo studies are warranted to validate the results of this study. Despite these limitations, our study elucidated the effects of senescence and SASP in PSCs on the progression of pancreatic cancer. This line of study would aid in the development of novel therapeutic agents for pancreatic cancer.

In conclusion, we clarified the promoting effects on pancreatic cancer cells and changes in gene expression profiles through senescence induction in hPSCs. Moreover, CXCLs were identified as SASP factors that promote proliferation and migration of pancreatic cancer cells. These promoting effects were attenuated by CXCR1/2 antagonist. A schematic representation is shown in Figure 6. Targeting senescent PSCs might be useful for the treatment of pancreatic cancer.

## 4. Materials and Methods

### 4.1. Ethics

The Ethics Committee of the Tohoku University Graduate School of Medicine approved the use of surgically resected human pancreatic cancer tissues (Article No. 2020-1-428). Written informed consent was obtained from all patients.

### 4.2. Materials

H_2_O_2_ and gemcitabine were purchased from Wako Pure Chemical Industries (Osaka, Japan). Rabbit antibody against p21 (#2947) were purchased from Cell Signaling Technology (Beverly, MA, USA). Mouse anti-p53 antibody (#sc-126) was purchased from Santa Cruz Biotechnology (Santa Cruz, CA, USA). Rabbit anti-GAPDH antibody (#2275-PC-100) was purchased from R&D Systems (Minneapolis, MN, USA). SB225002 (#S7651) and SCH-527123 (#S8506) were obtained from Selleck Chemicals (Houston, TX, USA). Other reagents were purchased from Sigma-Aldrich (St. Louis, MO, USA), unless otherwise specified.

### 4.3. Cell Culture

Human pancreatic cancer AsPC-1 and MIAPaCa-2 cell lines were obtained from the American Type Culture Collection (Manassas, VA). The cells were maintained in Dulbecco’s modified Eagle’s medium (DMEM) supplemented with 10% fetal bovine serum (FBS) and antibiotics. Primary hPSCs were isolated from the resected pancreatic tissues of patients who underwent surgery for pancreatic cancer, as previously described [55,56]. hPSCs were maintained in Ham’s F-12/DMEM (1:1) supplemented with 10% FBS and antibiotics in a humidified incubator with 5% CO_2_ at 37 °C. Experiments were performed using hPSCs at five to nine passages after isolation.

### 4.4. Treatment with H_2_O_2_ and Gemcitabine

Cellular senescence was induced by the treatment with H_2_O_2_ or gemcitabine. For the treatment with H_2_O_2_, hPSCs were left untreated or treated with H_2_O_2_ at the indicated concentrations for 2 h. After two washes with phosphate-buffered saline (PBS) (Wako Pure Chemical Industries), fresh medium supplemented with 10% FBS and antibiotics was added and cultured for 48 h [32]. In the case of gemcitabine treatment, hPSCs were cultured in the absence or presence of gemcitabine at the indicated concentrations for 48 h. For the preparation of conditioned media, senescence was induced with the treatments described above. Then, hPSCs were washed twice with PBS and cultured in Ham’s F-12/DMEM (1:1) supplemented with 1% FBS and antibiotics. After an additional 72 h, conditioned media were collected, centrifuged at 1700× *g* for 10 min, and sterilized using 0.22-μm filters (Merck Millipore, Billerica, MA, USA). Conditioned media were stored at −80 °C until use.

### 4.5. Assessment of SA-β-gal Activity

SA-β-gal activity was assessed using a senescence detection kit (#K320-250; BioVision, Milpitas, CA, USA) according to the manufacturer’s instructions. Briefly, hPSCs were seeded in a 12-well cultured plate (BD Biosciences, Franklin Lakes, NJ) and cultured overnight. The next day, the cells were washed with PBS and treated with H_2_O_2_ or gemcitabine. After 24 h, cells were washed with PBS and fixed for 15 min at room temperature in a fixative solution, followed by overnight incubation in a staining solution mix. SA-β-gal-positive cells stained with blue and negative cells were counted in five randomly selected fields of view at 200× magnification under a bright-field microscope, and the percentage of SA-β-gal-positive cells was calculated.

### 4.6. RNA Extraction and Quantitative Real-Time PCR

Total RNA was extracted using an RNeasy preparation kit (Qiagen, Valencia, CA, USA) according to the manufacturer’s instructions. Quantitative real-time PCR was performed using 1 μg of RNA and SuperScript VILO Master Mix (Thermo Fisher Scientific, Waltham, MA, USA). The expression of each gene was quantified using the StepOnePlus real-time PCR system (Thermo Fisher Scientific) and Fast SYBR Green Master Mix (Thermo Fisher Scientific) with the following primers: *CDKN1A*, forward 5′-TGTCCGTCAGAACCCATGC-3′, reverse 5′-CCAGTTGGTAACAATGCCATGT-3′ [57]; *TP53*, forward 5′-TAACAGTTCCTGCATGGGCGGC-3′, reverse 5′-AGGACAGGCACAAACACGCACC-3′ [58]; *ACTA2*, forward 5′-GTGTTGCCCCTGAAGAGCAT-3′; reverse 5′-GCTGGGACATTGAAAGTCTCA-3′; *COL1A1*, forward 5′-GAGGGCCAAGACGAAGACATC-3′, reverse 5′-CAGATCACGTCATCGCACAAC-3′; *COL4A1*, forward 5′-GGACTACCTGGAACAAAAGGG-3′, reverse 5′-GCCAAGTATCTCACCTGGATCA-3′; *COL5A1*, forward 5′-GCCCGGATGTCGCTTACAG-3′, reverse 5′-AAATGCAGACGCAGGGTACAG-3′; *CXCL1*, forward 5′-GATTGTGCCTAATGTGTT-3′, reverse 5′-ATCCAGATTGAACTAACTTG-3′; *CXCL2,* forward 5′-GGGCAGAAAGCTTGTCTCAA-3′, reverse 5′-GCTTCCTCCTTCCTTCTGGT-3′ [59]; *CXCL3*, forward 5′-CGCCCAAACCGAAGTCATAG-3′, reverse 5′-GCTCCCCTTGTTCAGTATCTTTT-3′ [60]; and *GAPDH*, forward 5′-GGCGTCTTCACCACCATGGAG-3′, reverse 5′-AAGTTGTCATGGATGACCTTGGC-3′ [12]. The mRNA expression levels were normalized to *GAPDH*.

### 4.7. Western Blotting

The cells were lysed in radioimmunoprecipitation buffer to obtain total protein. The samples were then electrophoresed using NuPAGE 8% Bis-Tris Gel (Thermo Fisher Scientific) and transferred onto Immobilon-P transfer membranes (Merck Millipore). Membranes were incubated overnight at 4 °C with primary antibodies. After incubation with peroxidase-conjugated secondary antibody for one hour at room temperature, the membranes were subjected to protein band visualization using ECL Western blot detection reagents (GE Healthcare, Buckinghamshire, UK). Densitometry analysis was performed using the ImageJ software (National Institutes of Health).

### 4.8. Microarray

Total RNAs were subjected to microarray analysis using a SurePrint G3 Human GE Microarray 8 × 60 K v3 (Agilent Technologies, Santa Clara, CA, USA). Data analysis was performed using the Linear Models for Microarray Analysis (limma) package [61] in the Bioconductor software [62]. To identify up- or downregulated genes, we calculated ratios (non-log-scaled fold change) from the normalized signal intensities of each probe for comparison between the control and H_2_O_2_-treated samples. The criteria for regulated genes were set as *P* < 0.05 and an absolute log-fold change (|logFC|) > 1.

### 4.9. GO and Pathway Enrichment Analyses

GO analysis is widely used to annotate large-scale genes and gene products [63]. GO was classified into three groups: BP, CC, and MF. KEGG is an encyclopedia of genes and genomes that is widely used to understand biological pathways and systems [64]. The Database for Annotation, Visualization, and Integrated Discovery (DAVID) (https://david.ncifcrf.gov/) (accessed on 23 June 2022) was used to perform GO and KEGG enrichment analyses of the differentially expressed genes of H_2_O_2_-treated hPSCs [65]. 

### 4.10. GSEA

We compared the gene profiles altered by senescence induction between hPSCs and hHSCs using GSEA [66]. The microarray dataset (GSE111954) of senescent hHSCs reported by Krizhanovsky et al. [29] was downloaded from the GEO database (https://www.ncbi.nlm.nih.gov/geo/) (accessed on 23 June 2022), and the Z-scores and ratios (non-log-scaled fold change) were calculated after quantile normalization. We established the following criteria for regulated genes: upregulated genes, Z-score ≥ 2.0, and ratio ≥ 1.5-fold; downregulated genes, Z-score ≤ −2.0 and ratio ≤ 0.66-fold. Next, we compared it with our array data of hPSCs using the limma package in the Bioconductor software [61,62].

### 4.11. ELISA

CXCL1 concentrations in conditioned media of hPSCs were measured using the Human CXCL1/GRO alpha Quantikine ELISA Kit (R&D Systems) according to the manufacturer’s instructions. The absorbance was measured at an optical density (OD) of 450 nm using Spectra Max M2e (Molecular Devices, Sunnyvale, CA, USA).

### 4.12. Cell Proliferation Assay

The proliferation of senescence-induced hPSCs was evaluated through direct cell counting. Senescence-induced hPSCs (5 × 10^3^ cells) were seeded in a 6-well plate (BD Biosciences) in Ham’s F-12/DMEM (1:1) supplemented with 10% FBS. After 48 and 96 h of incubation, cells were trypsinized and counted.

The proliferation of AsPC-1 and MIAPaCa-2 cells was assessed with a BrdU incorporation assay using the Cell Proliferation ELISA Kit (Roche Applied Science, Penzberg, Germany) according to the manufacturer’s instructions. AsPC-1 and MIAPaCa-2 cells (5 × 10^3^ cells/well) were seeded in a 96-well plate (BD Biosciences). After reaching 60–70% confluence, the cells were serum-starved overnight. The next day, cells were left untreated or treated with 10%H_2_O_2_ PSC-CM or GEM PSC-CM in the presence or absence of SB225002 or SCH-527123 at the indicated concentrations for 24 h. The cells were then labeled with BrdU for 2 h at 37 °C. The cells were fixed and incubated with a peroxidase-conjugated anti-BrdU antibody. Then, the peroxidase substrate was added, and BrdU incorporation was quantified with OD 370 to 492 nm using a Spectra Max M2e (Molecular Devices).

### 4.13. Cell Migration Assay

Cell migration was assessed by using wound-healing and two-chamber assays. In the wound-healing assay, AsPC-1 and MIAPaCa-2 cells were grown to confluence in a 6-well plate and incubated in a serum-starved medium overnight. The cell monolayer was mechanically scratched with a sterile 200-μL pipette tip. AsPC-1 and MIAPaCa-2 cells were left untreated or treated with 10% senescence-induced PSC-CM in the presence or absence of SB225002 (at 1 µM) or SCH-527123 (at 50 µg/mL). We used these reagents at these concentrations based on previous studies [67,68]. After 48 h, cell-free areas were measured using the ImageJ software (National Institutes of Health).

In the two-chamber assay, AsPC-1 (1 × 10^5^ cells/well) and MIAPaCa-2 (2 × 10^4^ cells/well) cells were seeded in the upper chamber with a pore size of 8 μm (BD Biosciences). The lower chamber included 10% senescence-induced PSC-CM in the presence or absence of SB225002 (at 1 µM) or SCH-527123 (at 50 µg/mL). After 48 h of incubation, cell migration towards the lower chamber was assessed. The cells in the upper chamber were removed with a cotton plug, and the cells that migrated through the membrane were fixed and stained with crystal violet. The stained cells were counted in five randomly selected fields under a bright-field microscope at 100 × magnification.

### 4.14. Statistical Analysis

The results are expressed as the mean ± standard deviation (SD). Experiments were performed at least three times, and similar results were obtained. Differences between two groups were assessed using Student’s *t*-test, whereas differences among multiple groups were assessed using the Tukey–Kramer method. We used JMP Pro 15 (SAS Institute Inc., Cary, NC, USA) for statistical analysis, and a two-sided *p*-value < 0.05 was considered to be statistically significant.

## Figures and Tables

**Figure 1 ijms-23-09275-f001:**
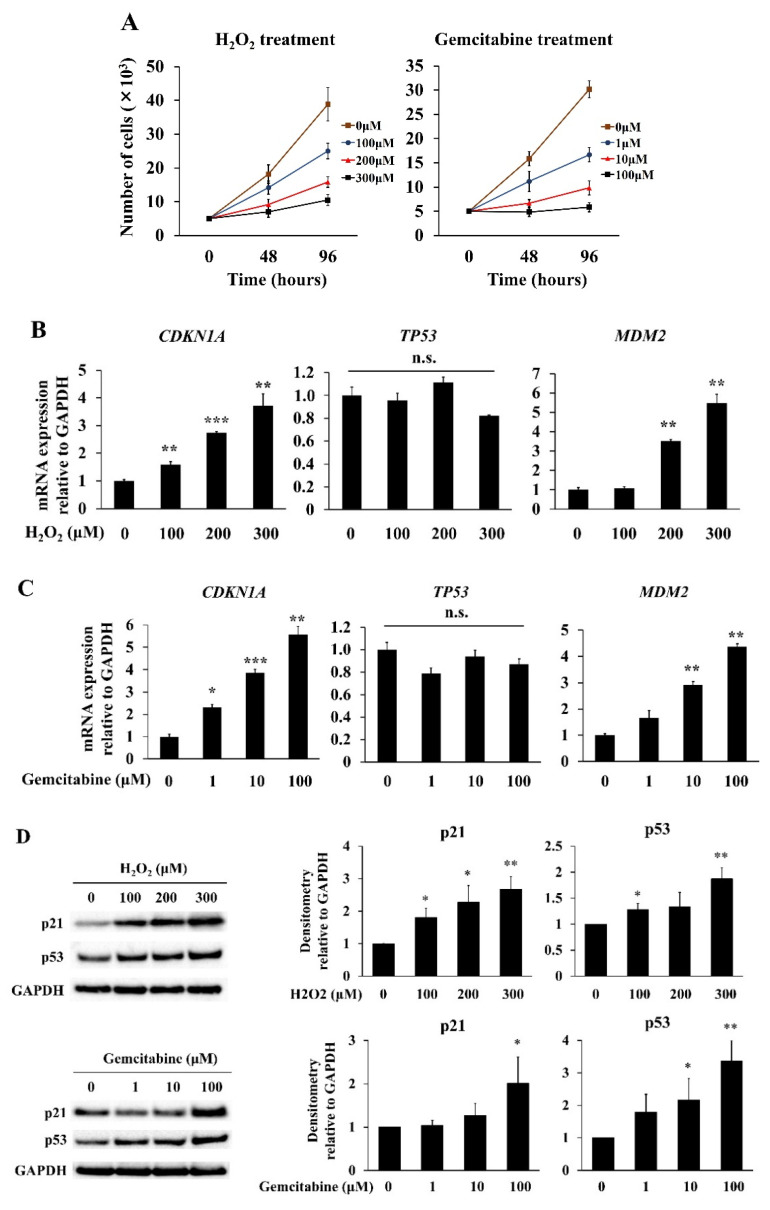
Hydrogen peroxide (H_2_O_2_)- and gemcitabine-induced senescence in hPSCs. Cellular senescence was induced in hPSCs through the treatment with H_2_O_2_ or gemcitabine at the indicated concentrations, as described in the Materials and Methods. (**A**) After 48 and 96 h of incubation, cells were trypsinized and counted. *n* = 5 each. (**B**,**C**) After 48 h of incubation, total RNAs were prepared, and the levels of *CDKN1A*/p21, *TP53*/p53, and *MDM2* mRNAs were determined through quantitative real-time PCR. The expression levels were normalized to GAPDH. *n* = 3 each. (**D**) After 48 h of incubation, the p21, p53, and GAPDH protein levels were determined by Western blotting. The expression levels of p21 and p53 relative to GAPDH in three independent experiments were quantified using the ImageJ software. (**E**,**F**) After 24 h of incubation, SA-β-gal staining was performed to detect senescent cells with blue. SA-β-gal-positive and negative cells were counted in five randomly selected fields at 200 × magnification. The percentages of SA-β-gal-positive cells are presented. Data are shown as the mean ± SD. SD bars are presented. * *p* < 0.05, ** *p* < 0.01, *** *p* < 0.001 vs. 0 μM. n.s., not significant.

**Figure 2 ijms-23-09275-f002:**
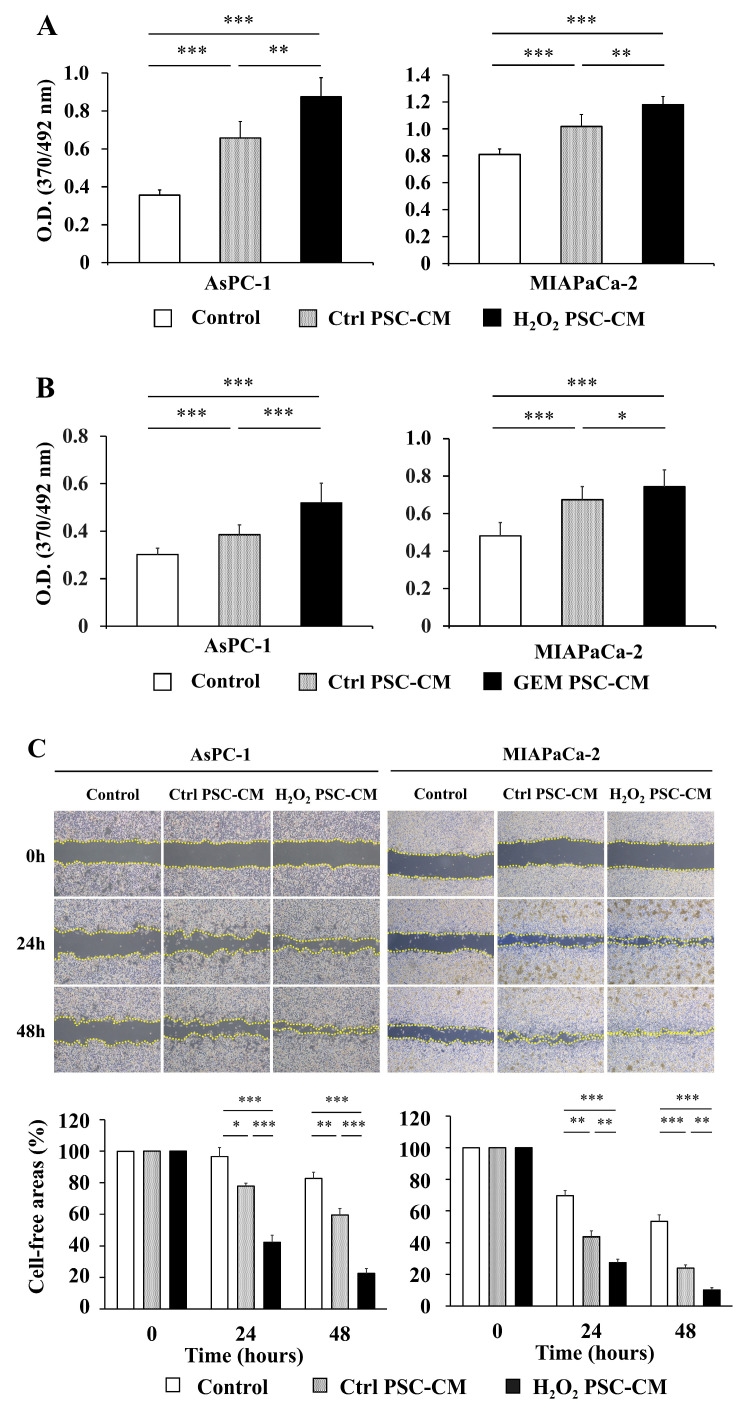
Conditioned media of senescence-induced hPSCs increased the proliferation and migration of pancreatic cancer cell lines. (**A**,**B**) AsPC-1 and MIAPaca-2 cells were left untreated (Control) or treated with 10% conditioned media of H_2_O_2_-treated hPSCs (H_2_O_2_ PSC-CM), gemcitabine-treated hPSCs (GEM PSC-CM), or untreated hPSCs (Ctrl PSC-CM) for 24 h. Cell proliferation was assessed with a BrdU incorporation assay. *n* = 10 each. (**C**,**D**) AsPC-1 and MIAPaca-2 cells were mechanically scratched after serum starvation overnight. Cells were left untreated (Control) or treated with 10% Ctrl PSC-CM, H_2_O_2_ PSC-CM, or GEM PSC-CM for 48 h. Cell-free areas were measured using the ImageJ software. *n* = 3 each from three independent experiments. (**E**,**F**) Cell migration was assessed with a two-chamber assay. Pancreatic cancer cells were seeded in a serum-free medium in the upper chamber, and the migration over 48 h toward the lower chamber containing 10% Ctrl PSC-CM, H_2_O_2_ PSC-CM, or GEM PSC-CM was evaluated. Migrated cells were counted in five randomly chosen high-power fields (100 × magnification). *n* = 15 each from three independent experiments. Data are shown as the mean ± SD. SD bars are presented. * *p* < 0.05, ** *p* < 0.01, *** *p* < 0.001. HPF, high-power field. n.s. = no significance.

**Figure 3 ijms-23-09275-f003:**
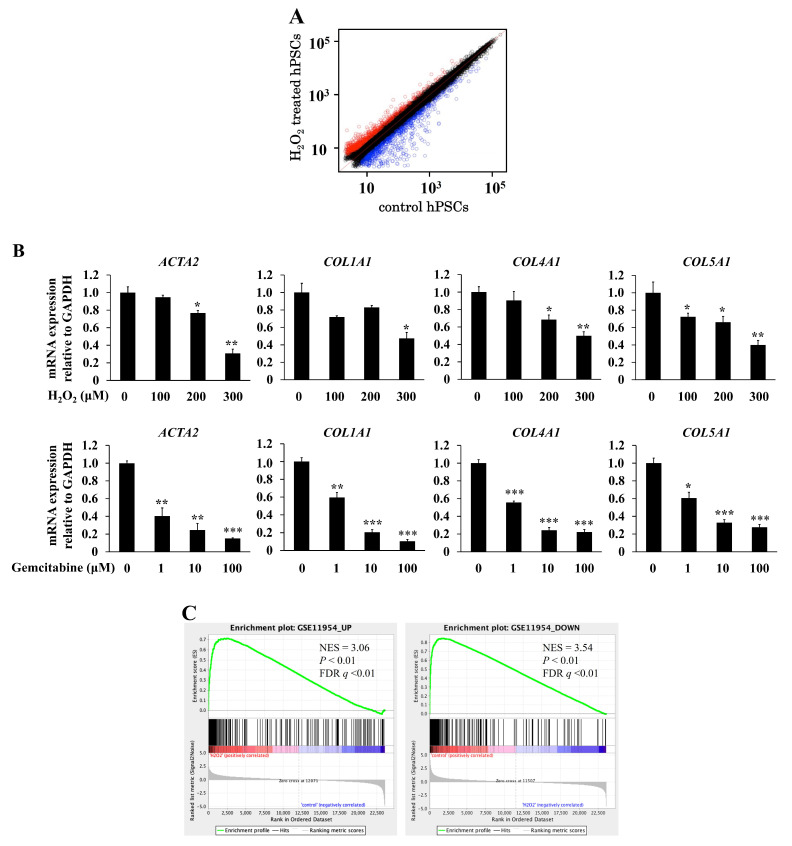
Gene expression profiles in senescence-induced hPSCs. Cellular senescence was induced in hPSCs with the treatment with H_2_O_2_ or gemcitabine at the indicated concentrations, as described in the Materials and Methods. (**A**) After 48 h of incubation, total RNAs were prepared and were subjected to Agilent’s microarray analysis. A scatter plot of the microarray is presented. X and Y axes are shown in Log_10_ scale. (**B**) After 48 h of incubation, total RNAs were prepared from hPSCs that were left untreated or treated with H_2_O_2_ or gemcitabine. The mRNA levels of actin alpha 2 (*ACTA2*/α-SMA), collagen type I alpha 1 chain (*COL1A1*), collagen type IV alpha 1 chain *(COL4A1*), and collagen type V alpha 1 chain (*COL5A1*) were determined through quantitative real-time PCR. The expression levels were normalized to GAPDH. Data are shown as the mean ± SD. SD bars are presented. * *p* < 0.05, ** *p* < 0.01, *** *p* < 0.001 vs. 0 μM (*n* = 3 each). (**C**) The upregulated (left panel) and downregulated (right panel) genes in the induction of senescence were compared between hPSCs and human hepatic stellate cells (hHSCs) using gene set enrichment analysis (GSEA). The gene set of senescence-induced hHSCs (GSE11954) was downloaded from the Gene Expression Omnibus database.

**Figure 4 ijms-23-09275-f004:**
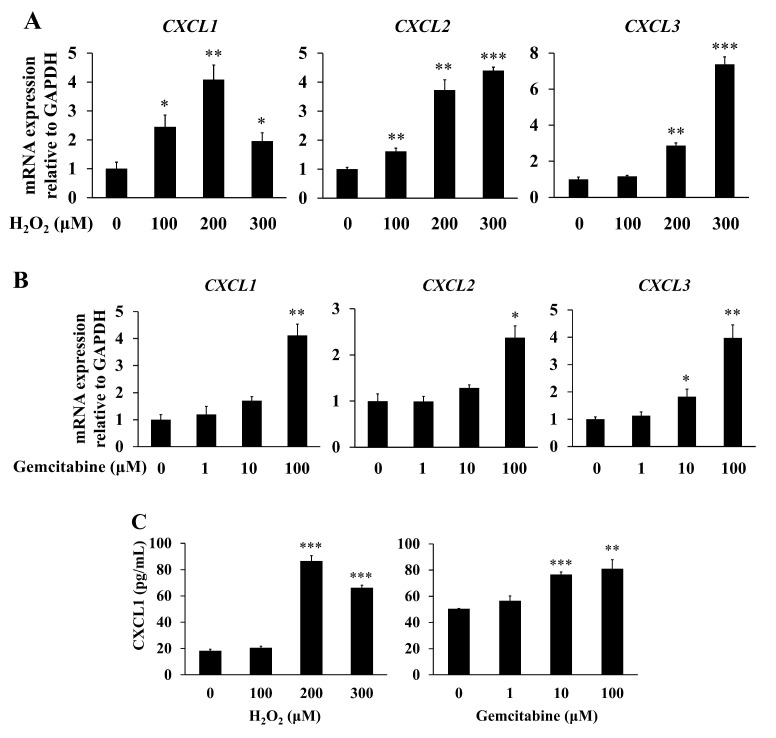
Senesce induction increased the expression of C-X-C motif chemokine ligand (CXCL)-1, CXCL2, and CXCL3 in hPSCs. hPSCs were treated with H_2_O_2_ or gemcitabine at the indicated concentrations, as described in the Materials and Methods. (**A**,**B**) After 48 h, total RNAs were prepared, and *CXCL1*, *CXCL2*, and *CXCL3* mRNA levels were determined through quantitative real-time PCR. The expression levels were normalized to GAPDH. *n* = 3 each. (**C**) After 48 h, conditioned media were prepared and the CXCL1 concentrations in the conditioned media were determined through ELISA. Data are shown as the mean ± SD. SD bars are presented. * *p* < 0.05, ** *p* < 0.01, *** *p* < 0.001 vs. 0 μM. *n* = 10 each.

**Figure 5 ijms-23-09275-f005:**
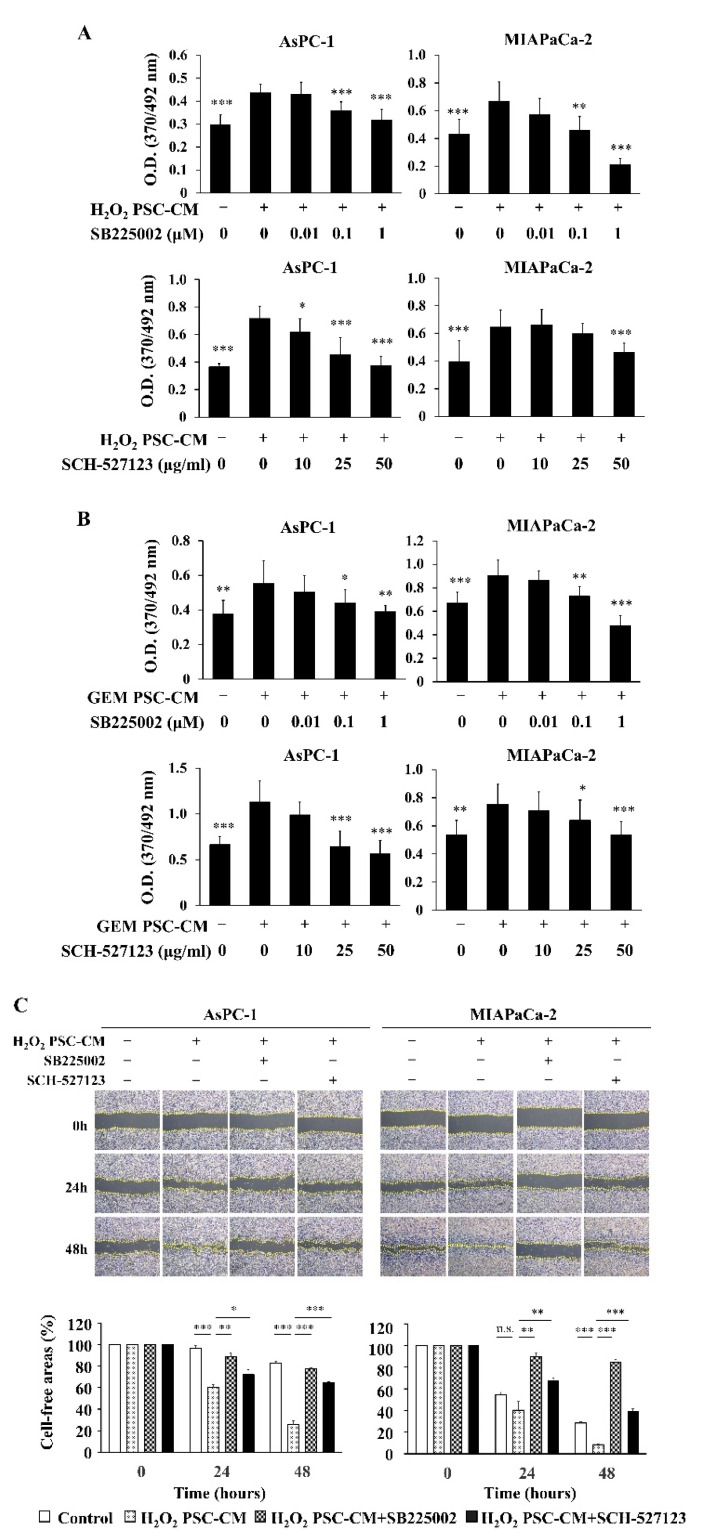
Inhibition of CXCR1/CXCR2 attenuates the proliferation and migration of pancreatic cancer cells induced by senescent hPSCs. Conditioned media were prepared from hPSCs treated with H_2_O_2_ (H_2_O_2_ PSC-CM) and those treated with gemcitabine (GEM PSC-CM), as described in the Materials and Methods. (**A**,**B**) AsPC-1 and MIAPaCa-2 cells were left untreated (Control) or treated with 10% H_2_O_2_ PSC-CM or 10% GEM PSC-CM in the absence or presence of a selective CXCR2 antagonist SB225002 at the indicated concentrations or a CXCR1/CXCR2 antagonist SCH-527123 at the indicated concentrations. After 24 h of incubation, cell proliferation was assessed with a BrdU incorporation assay. *n* = 10 each. (**C**,**D**) Serum-starved cells were mechanically scratched and left untreated or treated with 10% H_2_O_2_ PSC-CM or 10% GEM PSC-CM in the absence or presence of SB225002 at 1 μM or SCH-527123 at 50 μg/mL. After 48 h of incubation, cell migration was assessed by measuring cell-free areas using the ImageJ software. *n* = 3 each from three independent experiments. (**E**,**F**) Cell migration was also assessed with a two-chamber assay. AsPC-1 and MIAPaCa-2 cells were seeded in the upper chamber with a pore size of 8 μm. The lower chamber included 10% H_2_O_2_ PSC-CM or 10% GEM PSC-CM in the absence or presence of SB225002 at 1 μM or SCH-527123 at 50 μg/mL. After 48 h of incubation, cells that migrated toward the lower chamber were counted in five random fields under a bright-field microscope at 100 × magnification. *n* = 15 each from three independent experiments. Data are shown as the mean ± SD. SD bars are presented. * *p* < 0.05, ** *p* < 0.01, *** *p* < 0.001 vs. samples treated with the respective conditioned media without antagonists. n.s. = no significance.

**Figure 6 ijms-23-09275-f006:**
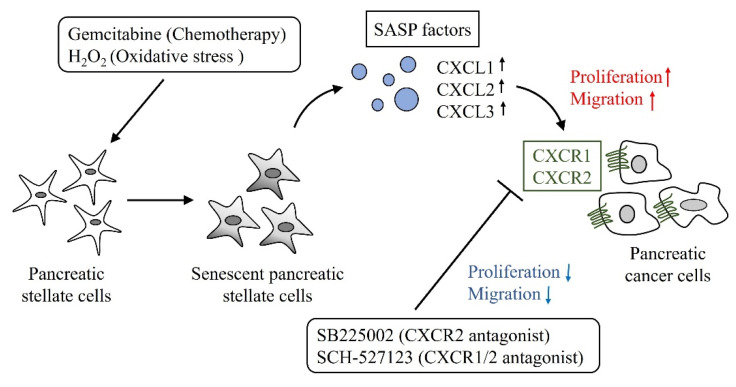
Schematic representation of this study. Gemcitabine, a standard drug for pancreatic cancer treatment, and H_2_O_2_, a reactive oxygen species, induce senescence in PSCs. Senescent PSCs secrete various SASP factors including CXCL1, CXCL2, CXCL3. These CXCLs, which act on CXCR1 and CXCR2, promote proliferation and migration of pancreatic cancer cells. SB225002, a CXCR2 antagonist, and SCH-527123, a CXCR1/CXCR2 antagonist, attenuate these cancer-promoting effects. H_2_O_2_, hydrogen peroxide. CXCL**,** C-X-C motif chemokine ligand. CXCR, C-X-C motif chemokine receptor.

**Table 1 ijms-23-09275-t001:** Top five Gene Ontology (GO) terms for senescence-induced hPSCs.

Category	Term	Gene Count	*p* Value
Biological process	Cell cycle	96	4.9 × 10^−33^
	Mitosis	58	1.4 × 10^−27^
	Cell division	68	5.9 × 10^−27^
	DNA replication	17	3.4 × 10^−7^
	Chromosome partition	8	9.3 × 10^−4^
Cellular component	Chromosome	76	1.2 × 10^−20^
	Kinetochore	29	9.8 × 10^−16^
	Centromere	33	1.7 × 10^−15^
	Microtubule	92	2.9×10^−8^
	Nucleosome core	34	7.1 × 10^−8^
Molecular function	Growth factor	19	2.0 × 10^−6^
	Cytokine	22	9.3 × 10^−6^
	Motor protein	15	5.7 × 10^−4^
	Protease inhibitor	14	7.6 × 10^−4^
	Heparin-binding	11	2.2 × 10^−3^

**Table 2 ijms-23-09275-t002:** Top 10 Kyoto Encyclopedia of Genes and Genomes (KEGG) pathways in senescence-induced hPSCs.

KEGG Pathway	Gene Count	*p* Value
Cell cycle	25	1.4 × 10^−10^
p53 signaling pathway	18	2.9 × 10^−9^
Alcoholism	22	1.9 × 10^−5^
Bladder cancer	10	2.8 × 10^−5^
Pathways in cancer	42	3.7 × 10^−5^
Focal adhesion	22	5.6 × 10^−5^
Small cell lung cancer	14	7.3 × 10^−5^
Systemic lupus erythematosus	17	1.1 × 10^−4^
Cytokine–cytokine receptor interaction	27	1.4 × 10^−4^
ECM–receptor interaction	13	1.9 × 10^−4^

## Data Availability

The data used in the current study are available from the corresponding author upon reasonable request.

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
