# Peer review of "Senescent Human Pancreatic Stellate Cells Secrete CXCR2 Agonist CXCLs to Promote Proliferation and Migration of Human Pancreatic Cancer AsPC-1 and MIAPaCa-2 Cell Lines"

_ijms, 2022, doi:10.3390/ijms23169275_

Round 1

Reviewer 1 Report

This is an interesting MS investigating the effects of conditioned medium (10%) from primarily cultured human pancreatic stellate cells made senescent by treatment with high concentrations of hydrogen peroxide (100 – 300 microM) and chemotherapeutic drug gemcitabine (1-100 microM). It was found that senescent human pancreatic stellate cells secrete endogenous CXCR2 agonists CXCL1-3, antagonism of CXCR2 with CXCR2 antagonist SB225002 or CXCR1/2 dual antagonist SCH-527123 blocked the effects of conditioned medium from senescent human pancreatic stellate cells on pancreatic cancer cell (AsPC-1, MIAPaCa-2) proliferation and migration. The data obtained were mostly well presented, but some concerns remain.

Major points

1/ Some statistical data from imaging experiments were from penta-plicates (5 random optical fields) out of a single experiment instead of multiple experiments.

2/ Some Western blots and the cell migration work (the cell wound assay) were not subject to quantitative analysis.

3/ These authors in studies of human tissues followed institutional guidelines for animal studies (Lines 284-286). 

Other points

4/ Title: The present title is rather unclear and cumbersome; the authors might want to change it to a title similar to “Primarily cultured senescent human pancreatic stellate cells secrete CXCR2 agonists CXCL1-3 to promote proliferation and migration of human pancreatic cancer cell lines AsPC-1 and MIAPaCa-2”.

5/ In each and every Figure legend and elsewhere, the authors claimed that “Error bars show the standard deviation.” The authors might have meant to say “Deviation bars are shown” or rather “Data are shown as mean plus / minus standard deviation, SD bars are presented”.

6/ Figure 1D: typical Western blots from a single experiment should be accompanied with quantitative data from multiples experiments.

7/ Section 2.2: Subtitle could be “Conditioned medium from senescent hPSC increases proliferation and migration of human pancreatic cancer cells”.

8/ Figures 2C, 2D, 5C, 5D: Typical images should be accompanied with quantitative data from multiple experiments (N > 3), possibly with blank or cell-free areas measured.

9/ Figure 5 legend: Title could be changed to “Inhibition of CXCR2 …”. You can inhibit a receptor; how could you inhibit an agonist?

10/ Figure 5: Only single dose of CXCR2 antagonist and CXCR1/2 dual antagonist were used. This should be explained, or please use multiple doses showing a clear dose-response relationship.

11/ Figures 1, 2, 5, Sections 4.5, 4.13: Cell senescence and migration was quantified by averaging 5 random optical fields from a single plate out of 1 singular experiment. Statistical data from multiple experiments should be presented.

12/ Authors should refrain from using too much non-standard abbreviations in the main text such as TME, GEM, … etc (which were probably used for everyday communications in the lab but are not suitable for written text).

13/ Discussion, Line 262-265: “... produce ECM proteins, cytokines, and growth factors. …. the inhibitory effects of PSC on pancreatic cancer have been reported”. Talking about senescence-associated secretory phenotype or rather diffusible bioactive factors released by pancreatic stellate cells, the authors might want to elaborate a bit more, in addition to ktheir present work about release of CXCL1-3 and their action via CXCR2 on target pancreatic cancer cells. It is known that bioactive factors such as cytokines released by pancreatic stellate cells could actually inhibit amylase secretion and cell surface receptor-mediated increases in cytosolic calcium concentration in pancreatic acinar cells (Figure 2C, Pancreatology 16: 570-577, 2016. doi: 10.1016/j.pan.2016.03.012; Figure 5, Cells 8: 109, 2019. doi: 10.3390/cells8020109).

14/ I suggest that Section 5 Conclusion move to the end of Discussion, as the last Para in Discussion. In the present MS, Discussion is without any conclusion.

15/ Line 292: Please indicate the Cat. No. for SB225002 and SCH-527123 as done for other major reagents used in this MS.

16/ Line 313: The centrifugal force should be shown in multiples of gravity g.

Reviewer 2 Report

The manuscript by Takikawa T and colleagues presents additional evidence supporting the tumor-promoting effects of senescent hPSCs, particularly through secreted CXCLs, on pancreatic cancer cells AsPC-1, and Mia PaCa-2. Manuscript is well-written and overall good presentation of the results. However, I have following queries:

Major:

Authors used AsPC-1 cells, which originates from ascites. I wonder, could authors have possibly used other pancreatic cancer cell lines such as BxPC-3 and Panc-1 instead of AsPC-1.

It would add value to the manuscript, if authors can provide some evidence on the impact of senescent hPSCs on chemosensitivity in pancreatic cancer cells.

Interestingly, authors only provide microarray data comparing ctrl vs H2O2 treated PSCs, while the data on GEM-treated hPSCs is missing. Can authors provide this data? In addition, the relevance of comparing hPSCs to hHSCs is unclear.  

Authors shall provide a detailed list/analysis of the hPSC secreted proteins with altered expression between ctrl PSC-CM and H2O2 PSC-CM or GEM PSC-CM.

How a single concentration of both SB225002 and SCH-527123 was chosen in unclear; is it the IC50 concentration that was used? Authors shall provide dose response data for both drugs in both cell lines used.

The conclusions of the study are based on a single hPSC, which is one of the major limitations of the study as the PSCs are known to differ phenotypically and in their interactions with pancreatic cancer cells, was recently shown by Lenggenhager D et al Cells 2019 (PMID: 30621293). I suggest author to mention this in the limitations and add ref.

Can authors provide a summary figure representing the finding of the study?

Minor:

Line 284-286: Ethics statement is misleading as it mentions animals use, however, there was no animal info or associated data is presented in the manuscript.

References used to described isolation of PSCs are old; authors may choose to add PMID: 31963309.

Fig. 1A: legends presented are mismatched between H2O2 and GEM conc, which does not match with the rest of Fig. 1. I suppose authors used H202= 100, 200 and 300 µM and GEM= 1, 10, 100 µM

Fig. 1B, C: change Y-axis label to “mRNA expression relative to GAPDH”

Fig. 2E, F: HPF is not defined in the figure text.

Fig. 3A: use simple label on X-axis e.g. control hPSCs or untreated hPSCs.

Fig. 3C, Fig. 4A-4C: Labels on X-axis are repeated four times, moreover, it is okay to use just GEM (µM) or H2O2 (µM). Change Y-axis label to “mRNA expression relative to GAPDH”

Fig. 5E, F: Bar diagram shall have similar colors to Fig. 5A, B, otherwise it is confusing for the reader

Table 1, 2: Last row is not necessary, as both terms are also described in the Table headings

It is better to avoid citing references in the Results section, and better to use them in the Discussion section.

Round 2

Reviewer 1 Report

I suggest that the authors make the following changes:

1/ Lines 224, 250: I suggest that the authors change "... senescence-induced ..." to "... senescent ...".

Author Response

Thank you for your careful review. We have revised the text according to your suggestion.

Reviewer 2 Report

Authors have addressed majority of my concerns and have implemented required changes in the text and figures in the revised manuscript. 

Author Response

Thank you again for your careful and constructive review, which contributed to the increased scientific quality of our paper.